# Phylogenetic Characterization of *Botryosphaeria* Strains Associated with *Asphondylia* Galls on Species of Lamiaceae

**Beata Zimowska** [1], **Sylwia Okoń** [2], **Andrea Becchimanzi** [3], **Ewa Dorota Krol** [1] **and Rosario Nicoletti** [3,4,*]

1 Department of Plant Protection, University of Life Sciences, 20-068 Lublin, Poland; beata.zimowska@up.lublin.pl (B.Z.); ewa.krol@up.lublin.pl (E.D.K.)
2 Institute of Plant Genetics, Breeding and Biotechnology, University of Life Sciences, 20-068 Lublin, Poland; sylwia.okon@up.lublin.pl
3 Department of Agricultural Sciences, University of Naples Federico II, 80055 Portici, Italy; andrea.becchimanzi@unina.it
4 Council for Agricultural Research and Economics, Research Centre for Olive, Citrus and Tree Fruit, 81100 Caserta, Italy
* Correspondence: rosario.nicoletti@crea.gov.it

**Abstract:** In the last decade, *Botryosphaeria dothidea* has been steadily reported as an associate of gall midges (Diptera, Cecidomyiidae) in a variety of host plants and ecological settings. This cosmopolitan fungus is well-known for its ability to colonize many plant species, as both a pathogen and an endophyte. Thus, the shift from this general habit to a lifestyle involving a strict symbiotic relationship with an insect introduces expectancy for possible strain specialization which could reflect separated phylogenetic lineages. Considering the recent taxonomic revision concerning species of *Botryosphaeria*, we evaluated the phylogenetic relationships among strains recovered from *Asphondylia* galls collected on several species of Lamiaceae in Poland and in Italy, and all the currently accepted species in this genus. A number of strains previously characterized from gall samples from Australia and South Africa, whose genetic marker sequences are deposited in GenBank, were also included in the analysis. As a result, full identity as *B. dothidea* is confirmed for our isolates, while strains from the southern hemisphere grouped separately, indicating the existence of genetic variation related to the geographic origin in the association with gall midges.

**Keywords:** *Asphondylia*; *Botryosphaeria*; *B. dothidea*; DNA sequencing; gall-associated fungi; Lamiaceae; phylogenetic relationships; symbiosis

## 1. Introduction

Although the nature of their symbiotic relationship has not been clearly ascertained, the occurrence of *Botryosphaeria dothidea* as an associate of many gall midge species (Diptera, Cecidomyiidae) is steadily reported, regardless of host plants and ecological contexts. For a long time, the identity of the fungal symbiont has been controversial, by reason of inherent difficulties in the isolation procedure, and of several taxonomic reassessments. In fact, several other fungi, such as *Cladosporium* spp. and *Alternaria* spp., have been frequently reported as gall associates, basically in connection with their saprophytic aptitude, which occasionally makes them conceal the real symbiont during the isolation procedure [1–4]. On the other hand, nomenclatural inconsistency, which only recently has been resolved after the epitypification of *B. dothidea* [5], may account for some previous incorrect

reports referring to *Macrophoma*, *Diplodia*, and *Dothiorella* [2,3,6,7]. Indeed, taxonomists recommend a careful interpretation of past literature concerning this fungus [8,9].

Like several species in the Botryosphaeriaceae, *B. dothidea* is well-known for its cosmopolitan distribution and ability to colonize a high number of plants, either as a pathogen or as an endophyte [8–10]. The involvement in cecidomyid-gall formation on a variety of plants confirms it as a very adaptive species. Although the shift from association with plants to a lifestyle characterized by a strict symbiotic relationship with an insect suggests possible strain specialization, which could reflect separated phylogenetic lineages, data resulting from previous studies did not provide evidence for this hypothesis [1,11]. However, in the last decade, the evolution in fungal taxonomy boosted by the application of DNA sequencing has provided a remarkable contribution in view of a better resolution of the *Botryosphaeria* species aggregate. Six species presenting a *Fusicoccum* anamorph, namely *B. corticis*, *B. dothidea*, *B. fabicerciana*, *B. fusispora*, *B. ramosa*, and *B. scharifii*, were recognized based on a detailed phylogenetic analysis; another species, *B. agaves*, is included in this group, although its anamorphic stage has never been described so far [8]. In addition to this basic set, several new species have been more recently identified, mostly based on isolations from tree plants in China (Table 1).

Species of Lamiaceae are widespread in the Mediterranean region, where they seem to represent a diversity hotspot for gall midges. In fact, two new species of the genus *Asphondylia* have been recently described from galls collected on host plants such as *Coridothymus capitatus* [12] and *Clinopodium nepeta* [4], and two more are in course of characterization from *Micromeria graeca* and *Clinopodium vulgare* (Viggiani, personal communication). However, their distribution appears to reach Central Europe, following the geographical spread of some hosts, as documented in the case of *A. serpylli* and *A. hornigi*, respectively associated with *Thymus* spp. and *Origanum vulgare* [3,13]. In the course of our investigations on a number of species of Lamiaceae, we had the opportunity to collect isolates from galls of *Asphondylia* spp. in two European countries with different climatic conditions. Despite a certain variation in biometric characteristics, sequence homology of internal transcribed spacers of ribosomal DNA (rDNA-ITS) confirmed *B. dothidea* as the fungal symbiont of gall midges in both contexts. However, the relative unreliability of data available in GenBank for this species, which is to be taken into account after the recent rearrangements within *Botryosphaeria*, prompted us to undertake a more accurate study of the phylogenetic relationships with the new taxa and other isolates from *Asphondylia* galls from other plant species/countries, in order to assess whether or not strains adapted to this particular symbiotic relationship are homogeneous in taxonomic terms.

**Table 1.** *Botryosphaeria* species with *Fusicoccum* anamorph described after 2013.

| Species | Main Hosts | Country | Reference |
|---|---|---|---|
| *B. auasmontanum* | *Acacia* | Namibia | [14] |
| *B. kawatsukai* | *Malus*, *Pyrus* | China | [15] |
| *B. minutispermatia* | dead wood | China | [16] |
| *B. pseudoramosa* | *Eucalyptus*, *Melastoma* | China | [17] |
| *B. qinlingensis* | *Quercus* | China | [18] |
| *B. qingyuanensis* | *Eucalyptus* | China | [17] |
| *B. rosaceae* | *Amygdalus*, *Malus*, *Pyrus* | China | [19] |
| *B. sinensia* | *Juglans*, *Malus*, *Morus*, *Populus* | China | [20] |
| *B. wangensis* | *Cedrus* | China | [17] |

## 2. Materials and Methods

### 2.1. Isolation and Morphological Observations

Isolations of fungal associates of gall midges were carried out on potato dextrose agar (PDA) amended with 85% lactic acid (1 mL·L$^{-1}$) in 90 mm diameter Petri dishes. After removing the outer residues of the flower calyx, fragments from gall walls were cut and transferred onto the agar medium. Plates were incubated in darkness at 25 °C. Hyphal tips from the emerging fungal colonies of the botryosphaeriaceous morphotype were transferred to fresh PDA plates, for morphological observations

and storage of pure cultures at 4 °C. Production of pycnidia was induced in cultures prepared in plates containing 2% water agar (WA), topped with sterilized pine needles, which were kept at room temperature under near-UV illumination [21]. Observations were carried out under a Motic BA 210 microscope (Xiamen, China), and images were taken through a 1 MP Motic camera and Scopelmage 9.0 software. Minimum, maximum, mean values, and the length/width (L/W) ratios of 50 conidia from each isolate were measured for a comparison with previously annotated reference sizes for *Botryosphaeria* species [8,17].

### 2.2. DNA Sequencing and Phylogenetic Analysis

Selected strains recovered from galls on several species of Lamiaceae collected in Italy and Poland (Table 2) were sampled from the surface of PDA cultures with a scalpel. The mycelial matter was transferred to 1.5 mL Eppendorf tubes, for DNA extraction. DNA isolation was performed by using a DNA easy plant and fungi isolation kit (EurX, Gdańsk, Poland), according to manufacturer's protocol. DNA concentration was estimated on 1.5% agarose gel, compared with GeneRulerTM DNA Ladder Plus (Thermo Scientific, Waltham, MA, USA), and measured by using a NanoDrop 2000 spectrophotometer (Thermo Scientific, Waltham, MA, USA). DNA samples were diluted to a concentration of 20 ng·µL$^{-1}$ and stored at 20 °C. Amplification of loci currently considered in taxonomy of *Botryosphaeria* [8] was carried out, using primers ITS1 and ITS4 for the rDNA-ITS region [22], and primers EF1-728F and EF1-986R for the translation elongation factor 1-alpha (TEF1) region [23]. PCR reaction mixtures contained 20 ng of genomic DNA, 0.2 mM of dNTP, 0.2 mM of each primer, $1 \times$ Taq buffer mM buffer (10 mM of Tris-HCl, 1.5 mM of $MgCl_2$, and 50 mM of KCl), and 1 U of Taq polymerase, and were adjusted to a final volume of 25 µL with sterile distilled water. PCR was conducted in a Biometra T1 thermocycler (Analytik Jena, Jena, Germany). The following reaction profile was applied: 95 °C—5 min, 35 cycles (95 °C—45 s, 52 °C—45 s, and 72 °C—45 s), with final elongation at 72 °C—5 min. PCR products were separated in 1.5% agarose gels containing EtBr in TBE buffer, at 140 V, for 1 h.

**Table 2.** Isolates of *B. dothidea* from *Asphondylia* galls collected on Lamiaceae used in this study.

| Number | Host | Location | GenBank Accession | |
|---|---|---|---|---|
| | | | ITS | TEF1 |
| SG3 | *Clinopodium nepeta* | San Giorgio a Cremano, Italy | MN731265 | MN737437 |
| AcE3 | *C. nepeta* | Astroni Nature Reserve, Italy | MN731266 | MN737438 |
| AcAs2 | *C. nepeta* | Astroni Nature Reserve, Italy | MN731267 | MN737439 |
| AcSe1 | *C. nepeta* | Serino, Italy | MN731268 | MN737440 |
| CLRi2 | *Clinopodium vulgare* | Rivello, Italy | MN731272 | MN737444 |
| Mp26j | *Mentha piperita* | Konopnica, Poland | MN731273 | MN737445 |
| Mp30p | *M. piperita* | Konopnica, Poland | MN731274 | MN737446 |
| MgBt1 | *Micromeria graeca* | Boscotrecase, Italy | MN731269 | MN737441 |
| MgPC6 | *M. graeca* | Palma Campania, Italy | MN731270 | MN737442 |
| MgPC7 | *M. graeca* | Palma Campania, Italy | MN731271 | MN737443 |
| OvdF3e | *Origanum vulgare* | Fajsławice, Poland | MN731275 | MN737447 |
| OvFs/g | *O. vulgare* | Fajsławice, Poland | MN731276 | MN737448 |
| Th/g2017 | *Thymus vulgaris* | Fajsławice, Poland | MN731277 | MN737449 |
| ThgI/10 | *T. vulgaris* | Fajsławice, Poland | MN731278 | MN737450 |

ITS: internal transcribed spacer; TEF1: translation elongation factor 1-alpha.

After checking and determining the size of the resulting PCR products, we submitted samples to Genomed (Warsaw, Poland), for sequencing. The obtained nucleotide sequences were compared with reference strains of *Botryosphaeria* spp. from GenBank. All sequences were checked and manually edited by using CLC Main Workbench 8.1.2 software (QIAGEN, Aarhus, Denmark) where necessary. Besides our original sequences, additional sequences of *Botryosphaeria* isolates from *Asphondylia* galls were searched in GenBank for inclusion in the phylogenetic analysis, where a strain of the species *Botryobambusa fusicoccum* was used as outgroup (Table 3). The combined and single ITS and TEF1 sequences were aligned by using Muscle [24] and manually adjusted with AliView software [25], where necessary. The phylogenetic analysis was conformed to a

recent protocol [26]. Congruence between the different datasets was tested by using the partition homogeneity test in PAUP software version 4.0b10 [27]. Gaps were treated as missing characters. Phylogenetic analyses of the concatenated and single-sequence data for maximum likelihood (ML) were performed by using RAxML software version 8.2.12 [28] with GTR+G model of nucleotide substitution and 1000 bootstrap replications. Concatenated sequences were also analyzed for maximum parsimony (MP) by using PAUP, under the heuristic search parameters with tree bisection reconnection branch swapping, 100 random sequence additions, maxtrees set up to 1000, and 1000 bootstrap. Posterior probabilities of the concatenated dataset were determined by Markov Chain Monte Carlo (MCMC) sampling in MrBayes version 3.0b4 [29]. MCMC chains were run for 4000,000 generations, sampling every 100, with a 25% burn-in discarded. Phylogenetic trees were drawn by using FigTree software [30]. Both the alignments and the trees of concatenated dataset were deposited in TreeBase (http://purl.org/phylo/treebase/phylows/study/TB2:S25558).

**Table 3.** Reference strains used in the phylogenetic analysis.

| Species | Number | GenBank Accession | |
| --- | --- | --- | --- |
| | | ITS | TEF1 |
| *Botryobambusa fusicoccum* | MFLUCC 11-0143 | JX646792 | JX646857 |
| *Botryosphaeria agaves* | MFLUCC 10-0051 | JX646790 | JX646855 |
| | MFLUCC 11-0125 | JX646791 | JX646856 |
| *Botryosphaeria auasmontanum* | CBS 121769 | KF766167 | EU101348 |
| | MFLUCC 15-0923 | MF398858 | MF398910 |
| | MFLUCC 17-1071 | MF398863 | MF398915 |
| *Botryosphaeria corticis* | CBS 119047 | DQ299245 | EU017539 |
| | ATCC 22927 | DQ299247 | EU673291 |
| *Botryosphaeria dothidea* | CBS 110302 | AY259092 | AY573218 |
| | CBS 115476 | AY236949 | AY236898 |
| | 3161 | EF614924 | EF614940 |
| | 3179 | EF614917 | EF614933 |
| | 3241 | EF614920 | EF614937 |
| | 3242 | EF614916 | EF614936 |
| | 3247 | EF614923 | EF614941 |
| | 3253 | EF614921 | EF614938 |
| | 3261 | EF614926 | EF614943 |
| | 3275 | EF614925 | EF614942 |
| | 3278 | EF614919 | EF614934 |
| | 3279 | EF614918 | EF614935 |
| *Botryosphaeria fabicerciana* | CMW 27094 | HQ332197 | HQ332213 |
| | CMW 27108 | HQ332200 | HQ332216 |
| *Botryosphaeria fusispora* | MFLUCC 10-0098 | JX646789 | JX646854 |
| | MFLUCC 11-0507 | JX646788 | JX646853 |
| *Botryosphaeria kawatsukai* | PGZH18 | MG637267 | MG637243 |
| | PGZH19 | MG637266 | MG637242 |
| *Botryosphaeria minutispermatia* | GZCC 16-0013 | KX447675 | KX447678 |
| | GZCC 16-0014 | KX447676 | KX447679 |
| *Botryosphaeria pseudoramosa* | CERC 2001 | KX277989 | KX278094 |
| | CERC 3455 | KX277997 | KX278102 |
| *Botryosphaeria qinlingensis* | CFCC 52984 | MK434301 | MK425020 |
| | CFCC 52985 | MK434302 | MK425021 |
| *Botryosphaeria qingyuanensis* | CERC 2946 | KX278000 | KX278105 |
| | CERC 2947 | KX278001 | KX278106 |
| *Botryosphaeria ramosa* | CBS 122069 | EU144055 | EU144070 |
| *Botryosphaeria rosaceae* | CFCC 82350 | KX197079 | KX197097 |
| | DZP B | KX197076 | KX197096 |
| *Botryosphaeria scharifii* | IRAN1529C | JQ772020 | JQ772057 |
| | IRAN1543C | JQ772019 | JQ772056 |
| *Botryosphaeria sinensia* | BJFU DZP141005-06 | KT343254 | KU221233 |
| | BJFU DZP141111-10 | KT343256 | KU221234 |
| *Botryosphaeria wangensis* | CERC 2298 | KX278002 | KX278107 |
| | CERC 2300 | KX278004 | KX278109 |

## 3. Results

Cultures on PDA of *Botryosphaeria* isolates recovered from *Asphondylia* galls on Lamiaceae displayed a sparse to moderately dense aerial mycelium, with diverse colors, from white-cream to gray to olivaceous-black, darkening with age, occasionally with narrow or wider columns of mycelium (Figure 1A). Pycnidial conidiomata developed after 10–14 days on PDA, or 8–12 days on pine needles in WA (Figure 1B). These fruiting bodies released buff, and, respectively, black (Figure 1C) or cream masses of spores containing typical *Fusicoccum* conidia, one-celled or with one septum (Figure 1D). They were smooth, hyaline, mostly with granular content, fusiform or irregularly fusiform, wider in the middle to upper third, base-truncate or subtruncate with rounded apex, and quite variable in size (Table 4). Muriform conidia referable to the synanamorphic stage *Dichomera* were never observed, unlike what previously resulted in subcultures of strains from galls collected on *T. vulgaris* directly prepared from the isolation plates [3]. This failure was assumed to possibly derive from prolonged storage at 4 °C of the stock cultures of our strains. Likewise, no isolate produced ascomata throughout the observation period.

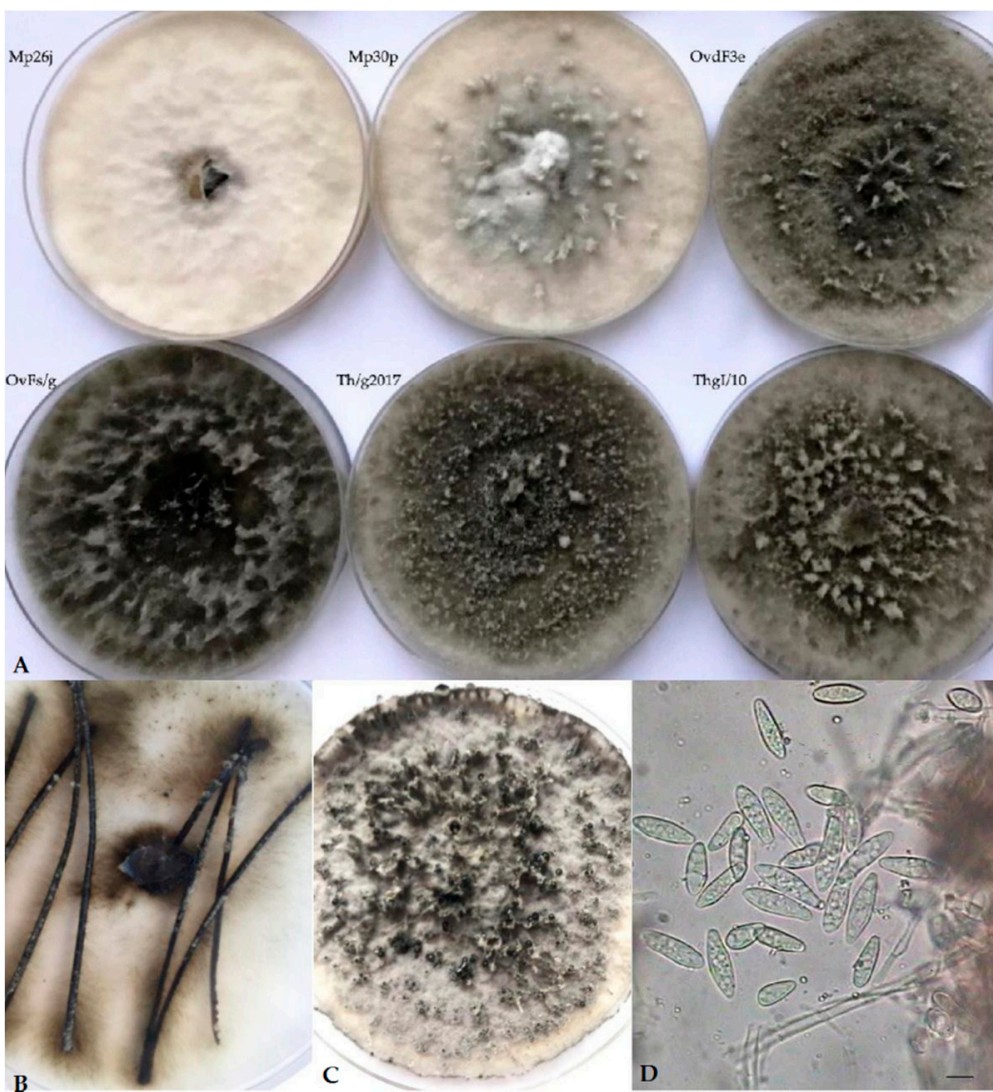

**Figure 1.** (**A**) Variable morphology of isolates from *Asphondylia* galls collected in our study. (**B**,**C**) Production of pycnidia, respectively, on pine needles on WA (water agar) and on PDA (potato dextrose agar). (**D**) *Fusicoccum* conidia.

**Table 4.** Morphological features of *B. dothidea* isolates from Lamiaceae examined in this study.

| Strain Number | Conidial Dimensions (Length × Width, µm) | | |
|---|---|---|---|
| | Range (50 Conidia) | Mean | Length/Width Ratio |
| SG3 | (14–)16.8–21.8 × (3.5–)4.2–5.6 | 18.9 × 4.9 | 3.2–5.2 |
| AcAs2 | 15.9–21 × (3.9–)4.3-5.8(–6.6) | 18.7 × 5.2 | 2.6–4.7 |
| AcE3 | (14.2–)16.4–21.6 × (3.4–)4.6–5.4 | 19.4 × 5.1 | 3.1–5.4 |
| AcSe1 | (13.4–)15.3–21(–22.9) × 3.8–5.7 | 18.2 × 5.1 | 2.3–5.1 |
| CLRi2 | 14.9–21.8 × (3.9–)4.3–5.8(–6.6) | 18.7 × 5.2 | 2.6–5.4 |
| Mp26j | (17.5–)18.5–29.7 × 3.7–7(–7.5) | 22.3 × 5.1 | 3.5–6.1 |
| Mp30p | (16.5–)19.0–28.7 × (3.2–)4.0–7.3 | 22.3 × 5.1 | 3.7–6.1 |
| MgBt1 | (13–)16.8-21.2 × 4.6–5.5(–6.2) | 17.3 × 5.2 | 2.4–4.6 |
| MgPC6 | (12.8–)15.3–21 × 3.8–5.7 | 15.6 × 4.9 | 2.1–5.5 |
| MgPC7 | (13.4–)15.5–21(–22.9) × 3.3–5.7 | 18.9 × 4.4 | 2.3–6.0 |
| OvdF3e | (17–)18.5–25.9 × (3.0–)3.5–7.4 | 24.3 × 5.3 | 3.0–6.1 |
| OvFs/g | (17.5–)18.5–28.7 × (2.8–)3.7–7.4 | 23.5 × 5.1 | 3.5–6.0 |
| Th/g2017 | (15.2–)16.5–25.7 × (2.7–)3.9–6.8 | 20.3 × 4.8 | 2.9–5.5 |
| ThgI/10 | (16.0–)18.5–28.4 × (3.1–)3.5–7.0 | 22.0 × 5.1 | 3.4–6.0 |

Because of the above unreliability of morphological characters, DNA sequence homology was fundamental for an accurate taxonomic identification. PCR products of approximately 560 bp for ITS region and 300 bp for TEF1 region were amplified and successfully sequenced in 14 isolates from *Asphondylia* galls considered in this study. All the nucleotide sequences obtained were deposited in GenBank (Table 2), and blasted against the ex-epitype strain of *B. dothidea* (CMW8000/CBS115476) [5]. Identity was 99.79% for all strains but one (MgBt1, 98.97%) for ITS sequences, while for TEF1 sequence identity was 100% except four isolates (Th/g2017, OvFs/g, OvdF3e, and again MgBt1) matching at 99.16%.

This identity was highlighted in the subsequent phylogenetic analysis considering reference strains of all the recognized *Botryosphaeria* spp. producing *Fusicoccum* conidia, along with previously identified strains of *B. dothidea* collected from *Asphondylia* galls, in other contexts, worldwide. Although several contributions have been published in recent years on the subject of gall midges and associated fungi, a search in GenBank showed that sequences of both ITS and TEF1 are only available for some isolates from *Acacia* spp., collected in Australia and South Africa, which were the subject of a previously mentioned study [1].

The trimmed and manually adjusted alignment of concatenated locus contained 58 strains (including the outgroup) and consisted of 490 and 255 bp for ITS and TEF1, respectively. The best scoring RAxML tree (Figure 2) had a final likelihood value of −1961.235515. The matrix had 172 distinct alignment patterns, with 5.74% of undetermined characters or gaps. The ML tree of ITS alone showed poor resolution compared to ML trees based on TEF1 and concatenated sequences (Figure S1). ML trees of TEF1 and ITS + TEF1 showed almost the same topology, except for *B. qingyuanensis*, which is included in the same clade of *B. wangensis*-*B. sinensis*-*B. qinlingensis* in the phylogram based on TEF1 alone (Figure S2). Parsimony analysis yielded 1000 equally parsimonious trees (tree length = 174 steps; consistency index = 0.902; retention index = 0.915; relative consistency index = 0.826; homoplasy index = 0.098). Of the 745 characters used, 76 were parsimony-informative, 65 were variable and parsimony-uninformative, and 604 were constant. The same clades were supported in MP and ML analyses, except for *B. minutispermatia*, which resulted in being less closely related to the Australian isolates in the MP tree (Figure S3).

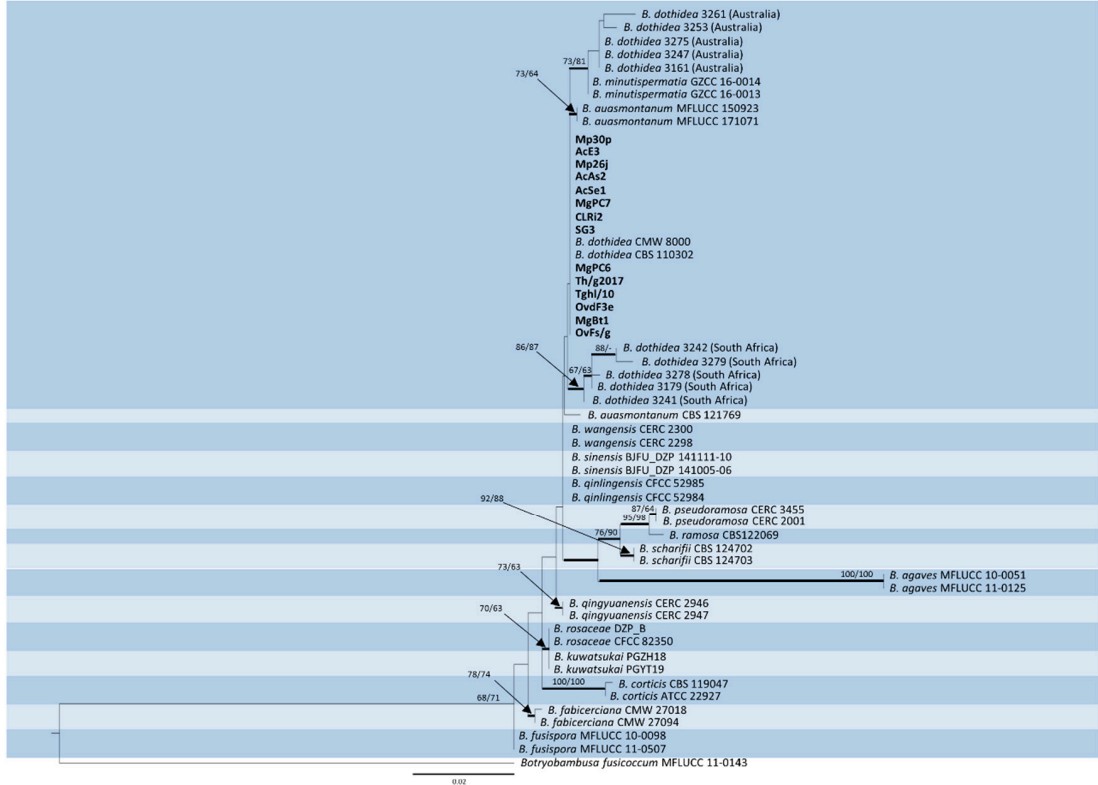

**Figure 2.** Phylogenetic tree based on maximum likelihood (ML) analyses of concatenated ITS and TEF1 sequences from strains considered in this study. Original isolates from Lamiaceae are in bold. Bootstrap support values ≥ 60% for ML and maximum parsimony (MP) are presented above branches as follows: ML/MP, bootstrap support values < 60% are marked with '-'. Branches in bold are supported by Bayesian analysis (posterior probability ≥ 90%). *Botryobambusa fusicoccum* MFLUCC 11-0143 was used as the outgroup reference.

## 4. Discussion

The wide morphological variation observed in our study is in line with recent findings that biometric data of *Botryosphaeria* species may overlap and are no longer relevant for the identification of *B. dothidea* [8,9,17]. Even *Dichomera* conidia are infrequently observed in this species [8,14,21], indicating that this character cannot be reliably taken into account for taxonomic purposes. Hence, reports from *Asphondylia* galls primarily referring to this anamorphic stage [1] should be considered with caution, because of the common co-occurrence of saprophytic *Alternaria* producing similar phaeodictyospores.

Therefore, nucleotide sequencing has become the primary identification method for *B. dothidea* through the assessment of homologies with the ex-epitype strain (CMW8000/CBS115476) [5], particularly considering the loci of ITS and TEF1. In fact, although ITS was shown to clearly distinguish *B. dothidea* from its closest relatives, the recent discovery of several cryptic species within the Botryosphaeriaceae makes it necessary to combine with TEF1 for a more reliable identification [8,31,32].

Phylogenetic analysis disclosed a clear identity with *B. dothidea* of our heterogeneous strain sample from Lamiaceae, regardless of their origin from two different climatic contexts. In fact, both Polish and Italian isolates grouped together with the type strains of this species. Conversely, Australian and South African strains formed two distinct groupings, which indicated a divergence from the most recent common ancestor. Particularly, the former was associated to type strains of the species *B. minutispermatia* in a more comprehensive clade, while the latter exhibited a slightly higher phylogenetic distance. Considering that both Australian and South African isolates were collected in association with gall midges on *Acacia* spp., this finding could be interpreted as being in agreement with a previous analysis

of the bulk *B. dothidea* sequences deposited in GenBank, showing a population structure which is shaped by geographical distance rather than host-plant preference [9]. In fact, lower identity with *B. dothidea* strains in the GenBank database resulted for these isolates, particularly those from South Africa whose TEF1 sequence homology was not higher than 96.83%. The observed variation particularly concerning this genetic marker requires further assessments to establish if these clusters should be interpreted as separated lineages within *B. dothidea*, or if they may represent distinct species.

The case of *B. auasmontanum*, a species characterized from a single strain (CBS121769) collected in Namibia [14], deserves further consideration. In fact, besides the holotype, our analysis included two more isolates from Italy ascribed to this species [26], whose ITS and TEF1 sequences are available in GenBank. Unexpectedly, they did not cluster with the holotype in the phylogenetic tree (Figure 2). A BLAST search in GenBank with sequences by these two strains clearly shows their 100% identity with *B. dothidea*, while ITS and TEF1 sequence homology with CBS121769 is lower (95.21% and 90.28%, respectively). The phylogenetic separation of the holotype of *B. auasmontanum* is supported by two large gaps resulting from the sequence alignment (Figure 3), which means that the claimed evidence of possible synonymy between the two species [26] applies only to the two Italian isolates. Thus, their inclusion in phylogenetic studies as representatives of *B. auasmontanum* should be avoided, and their incorrect taxonomic identification should be taken into account. Figure 3 also describes the substantial similarity among all strains of *B. dothidea*, including Australian and South African isolates, and the closely related *B. minutispermatia*, which essentially differ for nucleotide substitutions at single definite positions.

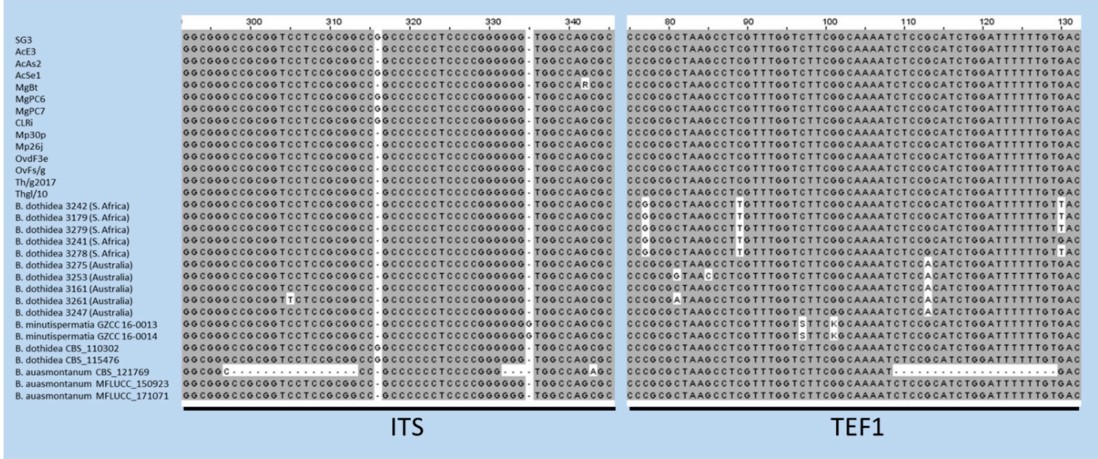

**Figure 3.** Alignment of ITS and TEF1 sequences of strains of *B. dothidea*, *B. minutispermatia*, and *B. auasmontanum*. Gaps in the sequences of the holotype of the latter species (CBS121769) are evident, along with single nucleotide substitutions in TEF1 sequences of Australian and South African isolates, and the type strains of *B. minutispermatia*.

## 5. Conclusions

Characterization of *Botryosphaeria* strains from different plant species and environmental contexts goes through the finding of novel species, with the expectation that additional undescribed taxa may be discovered in the future [17]. This concept involves strains associated with cecidomyids, inducing gall formation on a wide variety of plant species, which so far have been overlooked in assessments concerning taxonomy of such a highly adaptive and widespread fungus. Results of our investigation showed full identity with *B. dothidea* of isolates from galls collected from Lamiaceae, while a possible separation from this species should be verified for isolates previously recovered from *Acacia* in Australia and, particularly, South Africa. Indeed, a more adequate definition could be obtained by integrating these findings with data concerning cecidomyid-associated *Botryosphaeria* strains from other countries.

Therefore, a more active cooperation among researchers currently working on this topic worldwide is to be encouraged, in order to shed further light on this unique biological association.

**Supplementary Materials:** The following are available online at http://www.mdpi.com/1424-2818/12/2/41/s1. Figure S1: Maximum-likelihood tree of ITS sequence. Figure S2: Maximum-likelihood tree of EF sequence. Figure S3: One of the 1000 most parsimonious trees resulting from the analysis of concatenated ITS and TEF1 sequences.

**Author Contributions:** Conceptualization, B.Z. and R.N.; methodology, A.B., B.Z., E.D.K., and S.O.; formal analysis, A.B. and S.O.; writing—original draft preparation, B.Z., R.N., and S.O.; writing—review and editing, B.Z., E.D.K., and R.N.; funding acquisition, E.D.K. All authors have read and agreed to the published version of the manuscript.

**Funding:** This research received no external funding.

**Acknowledgments:** Authors thank Sarah Lucchesi (University of Southern Maine, Portland, USA) for revising the English style of this paper.

**Conflicts of Interest:** The authors declare no conflict of interest.

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
