# Peer review of "Phylogenetic Characterization of Botryosphaeria Strains Associated with Asphondylia Galls on Species of Lamiaceae"

_diversity, doi:10.3390/d12020041_

Round 1
Reviewer 1 Report
Zimowska et al. present a relevant report of a novel genetic variant of Botryosphaeria on Lamiaceae hosts. The methods and procedures used in the manuscript are solid and are very relevant for the plant pathology community.
However, there are major issues with the way the authors communicate their ideas, mainly in the discussion. The language is convoluted and very complicated in some parts. I understand that English may not be the main language of the authors and how complicated simple, scientific English can be. But in this case, its necessary that the authors rewrite parts of the discussion in order to make this manuscript accessible to the community. These results are very relevant for the understanding of the diversity of this genus, and clear, simple and concise writing can make this manuscript a fundamental read for the plant pathology community interested in Botryosphaeria diversity.
Specific comments and recommendations are included in the attached file.

Author Response
Zimowska et al. present a relevant report of a novel genetic variant of Botryosphaeria on Lamiaceae hosts. The methods and procedures used in the manuscript are solid and are very relevant for the plant pathology community.
However, there are major issues with the way the authors communicate their ideas, mainly in the discussion. The language is convoluted and very complicated in some parts. I understand that English may not be the main language of the authors and how complicated simple, scientific English can be. But in this case, its necessary that the authors rewrite parts of the discussion in order to make this manuscript accessible to the community.
Thank you for your favorable comments and indications for improving our paper. The text has been revised by an English mother tongue expert, and we hope the new version meets your requirements for a higher accessibility. We have also adapted the text according to your specific comments. In this respect, would you please note that Genomed is name of the laboratory providing the sequencing service, while accession numbers of reference strains have been provided in Table 3.
These results are very relevant for the understanding of the diversity of this genus, and clear, simple and concise writing can make this manuscript a fundamental read for the plant pathology community interested in Botryosphaeria diversity.
Specific comments and recommendations are included in the attached file.
Finally, concerning your comment at line 205 we decided to be cautious in separating a new species at the current stage, considering the opportunity to integrate our findings with additional data from other contexts worldwide, as it is prompted in the conclusions.
Reviewer 2 Report
Dear Editor,
I recommend the manuscript for publication after some adjustments have been taken into account. All the revisions are marked in detail in the attached file.

Author Response
Dear Editor,
I recommend the manuscript for publication after some adjustments have been taken into account. All the revisions are marked in detail in the attached file.
Thank you for your positive judgement, and suggestions for improving our manuscript. The new version was integrated in order to incorporate your observations, particularly concerning the phylogenetic analysis. We performed a single locus analysis (ITS and TEF1), which is now presented in the supplementary materials. As for comments on the isolation procedure, we specify that it was quite easy for us to recognize fungi of the botryosphaeriaceous morphotype at an early stage of growth in the isolation plates after performing hundreds of isolations from Asphondylia galls, and that galls were not previously sterlized in order not to impair the isolation outcome. Moreover, it must be considered that galls are contained in the calyx residues which are to be removed before dissection, and at some extent represent a protective layer against external contaminants.
Reviewer 3 Report
This paper reports a bunch of new isolates of B. dothidea, together with their ITS and TEF sequences. To me, it is unclear what were the aims of the study and what should be its take-home message. The whole paper needs to be rewritten in a way that the take-home message will be clear. Perhaps, additional data could be needed in order to reach the status of the minimal publishable unit. Detailed comments follow:
Introduction – specify the aims and/or hypotheses at the end of the Introduction. Methods – specify the exact agar composition; specify manufacturers of all chemicals used 97 – typo “NanoDro” Table 3 – provide references for sequences that did not originate from this study. 2 – NCBI accession numbers should be provided for all the isolates. 2 – there is very limited number of informative bases when attempting to differentiate between dothidea and, e.g., aquasmontanum. There is clearly no space left for the identification of intraspecific variability. The ITS and TEF sequences are ok for the species identification in the examined genus. However, to address the intraspecific variability, additional markers would need to be examined. To allow the assessment of proper alignment of the sequences, their alignment needs to be submitted as a supplementary file. 2 – bootstrap support for the most problematic nodes is not shown The interpretation of Fig. 2 with respect to aquasmontanum and minutispermatia lacks sufficient support based on the provided data. I would avoid publishing any such conclusions as they seem to be too premature. The conclusion might be correct but you simply cannot claim that based on the available evidence. Recently, other fungi, including Penicillium and Aspergillus spp. were reported to be able to infect insects in galls made by other gall midge species on common reed – see J. Invertebr. Pathol. 133 (2016): 95-106 and Fung. Div. 66 (2014): 89-97 – it would be good to comment on whether the method used (PDA agar inoculation) would allow their growth and what could be the reasons of their absence in the examined. Similarly to Botryosphaeria, I would consider those species to be rather saprophytes and was surprised to read that they can colonize living larvae.Author Response
This paper reports a bunch of new isolates of B. dothidea, together with their ITS and TEF sequences. To me, it is unclear what were the aims of the study and what should be its take-home message. The whole paper needs to be rewritten in a way that the take-home message will be clear. Perhaps, additional data could be needed in order to reach the status of the minimal publishable unit.
Thank you for your contribution to the improvement of our paper, which we have modified in order to incorporate your observations.
Detailed comments follow:
Introduction – specify the aims and/or hypotheses at the end of the Introduction. Methods – specify the exact agar composition; specify manufacturers of all chemicals used 97 – typo “NanoDro” Table 3 – provide references for sequences that did not originate from this study. 2 – NCBI accession numbers should be provided for all the isolates. 2 – there is very limited number of informative bases when attempting to differentiate between dothidea and, e.g., aquasmontanum.
Particularly, we have better specified the aims in the introduction, and checked correctness of accession numbers of sequences, while references for them are provided in table 1, or elsewhere in the text.
There is clearly no space left for the identification of intraspecific variability. The ITS and TEF sequences are ok for the species identification in the examined genus. However, to address the intraspecific variability, additional markers would need to be examined.
We observe that the issue of intraspecific variability has been addressed for B. dothidea (upper part of the tree), while this is not the case for the other species; however, the latter just represent a reference according to what resulted from the phylogenetic analysis.
To allow the assessment of proper alignment of the sequences, their alignment needs to be submitted as a supplementary file. 2 – bootstrap support for the most problematic nodes is not shown The interpretation of Fig. 2 with respect to aquasmontanum and minutispermatia lacks sufficient support based on the provided data. I would avoid publishing any such conclusions as they seem to be too premature.
We cannot add a supplementary file displaying alignment of sequences because of their high number; however the alignment can be checked at http://purl.org/phylo/treebase/phylows/study/TB2:S25558?x-access-code=27f3f8b8028c144c669b10ebf7ab6866&format=html, where it is uploaded as nexus file along with the correspective tree.
The conclusion might be correct but you simply cannot claim that based on the available evidence. Recently, other fungi, including Penicillium and Aspergillus spp. were reported to be able to infect insects in galls made by other gall midge species on common reed – see J. Invertebr. Pathol. 133 (2016): 95-106 and Fung. Div. 66 (2014): 89-97 – it would be good to comment on whether the method used (PDA agar inoculation) would allow their growth and what could be the reasons of their absence in the examined. Similarly to Botryosphaeria, I would consider those species to be rather saprophytes and was surprised to read that they can colonize living larvae.
Finally, we confirm that, unlike Alternaria and Cladosporium, we have never happened to isolate Penicillium or Aspergillus from the galls. This was surprising for us too, considering that in our research activity we have often had the opportunity to recover strains of these fungi from both terrestrial and marine contexts.
Round 2
Reviewer 3 Report
The revision only poorly reflected the provided comments. The study failed to identify the research gap, which would be worth to be addressed by the chosen approach.
Author Response
The revision only poorly reflected the provided comments. The study failed to identify the research gap, which would be worth to be addressed by the chosen approach.
Indeed this comment is quite vague, and frankly we do not know how any our further modification could be satisfactory. At the same time, we consider quite meaningful that reviewers 1 and 2 did not identify any 'research gap' that we may have failed to address, and that it seems they were happy with the revised version we submitted in response to the first round of revision.